# The computational and energy cost of simulation and storage for climate science: lessons from CMIP6

Mario C. Acosta[1], Sergi Palomas[1], Stella Paronuzzi[1], Gladys Utrera[1], Joachim Biercamp[2], Pierre-Antoine Bretonniere[1], Reinhard Budich[3], Miguel Castrillo[1], Arnaud Caubel[4], Francisco Doblas-Reyes[1], Italo Epicoco[5], Uwe Fladrich[6], Sylvie Joussaume[4], Alok Kumar Gupta[7], Bryan Lawrence[8], Philippe Le Sager[9], Grenville Lister[8], Marie-Pierre Moine[10], Jean-Christophe Rioual[11], Sophie Valcke[10], Niki Zadeh[12], and Venkatramani Balaji[13]

[1]Barcelona Supercomputing Center, Plaça d'Eusebi Güell, 1-3, 08034 Barcelona, Spain
[2]German Climate Computing Centre, Bundesstraße 45a, 20146 Hamburg, Germany
[3]Max Planck Institute, Hofgartenstr. 8, 80539 Munich, Germany
[4]Institut Pierre-Simon Laplace, 11 Bd d'Alembert, 78280 Guyancourt, France
[5]Euro-Mediterranean Center on Climate Change, Via della Libertà, 12, 30121 Venezia VE, Italy
[6]Swedish Meteorological and Hydrological Institute, SE-601 76 Norrköping, Sweden
[7]Norwegian Research Centre, Nygårdsgaten 112, 5008 Bergen, Norway
[8]National Centre for Atmospheric Science, Fairbairn House, 71-75 Clarendon Rd, Woodhouse, Leeds LS2 9PH, United Kingdom
[9]Royal Netherlands Meteorological Institute, Utrechtseweg 297. NL-3731 GA De Bilt, Netherlands
[10]European Center for Advanced Research and Training in Scientific Computing, 42 Av. Gaspard Coriolis, 31100 Toulouse, France
[11]Meteorological Office, Fitzroy road, Exeter, Devon, EX1 3PB, United Kingdom
[12]National Oceanic and Atmospheric Administration, 1401 Constitution Avenue NW, Room 5128, Washington, DC 20230, USA
[13]High Meadows Environmental Institute, Princeton University, Guyot Hall, Room 129, Princeton, NJ 08544-1003, USA

**Correspondence:** Mario C. Acosta (mario.acosta@bsc.es), Sergi Palomas (sergi.palomas@bsc.es)

**Abstract.** The Coupled Model Intercomparison Project (CMIP) is one of the biggest international efforts aimed at better understanding past, present, and future of climate changes in a multi-model context. A total of 21 Model Intercomparison Projects (MIPs) were endorsed in its 6th phase (CMIP6), which included 190 different experiments that were used to simulate 40000 years and produced around 40 PB of data in total. This paper presents the main findings obtained from the CPMIP (the Computational Performance Model Intercomparison Project) a collection of a common set of metrics, specifically designed for assessing climate model performance. These metrics were exclusively collected from the production runs of experiments used in CMIP6 and primarily from institutions within the IS-ENES3 consortium. The document presents the full set of CPMIP metrics per institution per experiment, including a detailed analysis and discussion of each of the measurements. During the analysis, we found a positive correlation between the core hours needed, the Complexity of the models, and the Resolution used. Likewise, we show that between 5-15% of the execution cost is spent in the coupling between independent components, and it only gets worse increasing the number of resources. From the data, it is clear that queue times have a great impact on the actual speed achieved, and have a huge variability across different institutions, ranging from none to up to 78% execution

overhead. Furthermore, we evaluated that the estimated carbon footprint of running such big simulations within the IS-ENES3 consortium is 1692 tons of CO2 equivalent.

As a result of the collection, we contribute to the creation of a comprehensive database for future community reference, establishing a benchmark for evaluation and facilitating the multi-model, multi-platform comparisons crucial for understanding climate modelling performance. Given the diverse range of applications, configurations, and hardware utilised, further work is required for the standardisation and formulation of general rules. The paper concludes with recommendations for future exercises aimed at addressing the encountered challenges which will facilitate more collections of a similar nature.

*Copyright statement.*  TEXT

## 1   Introduction

Earth System Models (ESMs) are an essential tool for understanding the Earth's climate and the consequences of climate change, which are crucial to the design of response policies to address the current climate emergency resulting from anthropogenic emissions. Modelling the Earth is inherently complex. ESMs are among the most challenging applications that the

25 High-Performance Computing (HPC) industry has had to face, requiring the most powerful computers available, consuming vast amounts of energy in computer power, and producing massive amounts of data in the process (Wang and Yuan 2020; Wang et al. 2010; Fuhrer et al. 2014; McGuffie and Henderson-Sellers 2001; Dennis et al. 2012).

    Virtually all models are designed to exploit the parallelism of HPC machines so that we can get the results in a reasonable amount of time while trying to make the best use of the HPC platform. While the technology underneath keeps improving

every year (in Petaflops/s, memory bandwidth, I/O speed, etc.) climate software evolves much more slowly. Balaji (2015) and Liu et al. (2013) show how challenging is to adapt multi-scale multi-physics climate models to new hardware or programming paradigms. These models, often community-developed software, are very complex, inherently chaotic, and subject to numerical stability. All of which contribute to a slower evolution of the codes. Bauer et al. (2021) illustrate how climate science did take advantage of Moore's law (Bondyopadhyay, 1998) and Dennard scaling (Frank et al., 2001) without much

pressure to fundamentally revise numerical methods and programming paradigms, leading to huge legacy codes mostly driven by scientific concerns. Consequently, such codes achieve notably poor sustained floating-point performance in present-day CPU architectures. Enhancing the performance of these models is crucial to boost the rate at which they can grow (in the resolution, complexity, and features represented). In a context where energy cost is rising, running faster and more cost-effective simulations is key to contributing to the advancement of climate research.

The performance of ESMs is hardly limited only by one but by multiple bottlenecks that depend on the model itself and on the properties of the HPC platform on which they run, for instance: models using higher resolutions may benefit from (or be limited by) the speed of the network as the data is split into many nodes, memory-bound models will benefit from having more memory available per core and with faster transmission speed while compute-bound models will perform better in faster CPUs,

models that produce more output will run faster on infrastructures with higher capacities for I/O operations, models that include more individual components will be limited by the load-balance achieved between them and by the coupler performance.

Balaji et al. (2017) proposed a set of 12 performance metrics that define the Computation Performance for Model Intercomparison Project (CPMIP) which were specifically designed for climate science by taking into account the structure of ESMs and how they are executed in real experiments. This set of metrics include the climate experiment and platform properties, the computational speed and cost (core-hours and energy), measures for the coupling and I/O overhead, and for the memory consumption. Each one is described in detail in Table 1 and Section 3.

In this paper, we present in Section 2 the collection of CPMIP metrics from 33 experiments used for climate projections in the Coupled Model Intercomparison Project phase 6 (CMIP6). The collection effort has been predominantly led by institutions affiliated with the IS-ENES3 (Joussaume, 2010), a consortium founded by a Horizon 2020[1] project composed of the most important weather and climate centres in Europe and devoted to improving the infrastructure to make the Earth System Grid Federation (ESGF) and CMIP publication easier. This compilation is the first of its kind and constitutes a representative part of the total 124 CMIP6 experiments, involving 45 institutions[2]. Our data encompasses 33 different experiments that were used to simulate almost 500 000 years during CMIP6 on 14 different HPC machines and involving 14 independent modelling institutions. All experiments are listed in Table 2, along with the institution in charge, the experiment name, HPC platform, ocean and atmosphere resolutions, and the main reference to the experiment configuration. In addition, Table 3 shows the complete collection of CPMIP metrics for each one of the models, and Table 4 lists the HPC machines that have been used to run these models. Furthermore, in Section 3, we include the analysis of the metrics to underscore the most significant insights derived from this data collection. We study in detail the measurements reported by each institution, grouping them based on experiment configurations, establishing relationships between intertwined metrics, and discussing the strengths and difficulties encountered during the analysis of each metric. For instance, our analysis reveals that institutions tend to increase the number of resources used in higher-resolution experiments, thereby mitigating the expected increase in execution time at the expense of increasing the core-hours required. Similarly, the addition of extra components simulated increases the core-hours needed, and the cost of coupling interactions and synchronisations between models as well. Institutions reported that the Coupling cost entails an execution cost overhead typically ranging between 5-15%, and it tends to be more problematic higher processor counts. Additionally, the numbers indicate that the volume of data generated by an experiment does not correspond to increases in Resolution or core-hours needed, contrary to expectations. We observed very different queue times for HPC resources across institutions, ranging from instantaneous access to introducing an execution time overhead of up to 78%. Furthermore, we present an initial approximation of the carbon footprint generated from executing these experiments, totaling 1692 tons of CO2 equivalent.

Our study emphasises the significance of developing standardised metrics for assessing climate model performance. This contribution will serve to establish a database for future reference and multiple institutions-modellers will be able to use for

---

[1]https://research-and-innovation.ec.europa.eu/funding/funding-opportunities/funding-programmes-and-open-calls/horizon-2020_en, retrieved February 6, 2024.

[2]http://esgf-ui.cmcc.it/esgf-dashboard-ui/data-archiveCMIP6.html, retrieved February 6, 2024.

**Table 1.** List of CPMIP metrics collected

| Metric | Used to evaluate |
|---|---|
| Resolution (Resol) | number of grid points NXxNYxNZ per component |
| Complexity (Cmplx) | number of prognostic variables per component |
| Platform | machine measurements: core count, clock frequency, and double-precision op. per clock cycle |
| Simulation Years Per Day (SYPD) | number of simulated years per day (24h) of execution time |
| Core-hours per Simulated Year (CHSY) | cost, measured in core-hours per simulated year |
| Actual SYPD (ASYPD) | how queue time and interruptions affect the complete experiment duration |
| Parallelisation (Paral) | total number of cores allocated for the run |
| Joules Per Simulated Year (JPSY) | energy needed per year of simulation |
| Memory Bloat (Mem B) | ratio between actual and ideal memory size |
| Data Output cost (DO) | computing cost for performing I/O |
| Data Intensity (DI) | amount of data produced after 1 year of simulation divided by the CHSY |
| Coupling Cost (Cpl C) | computing cost of the coupling algorithm and load imbalance |

comparison, which we believe to be essential for evaluating climate modeling performance. The noise and variability present in the dataset are the results of the diversity of the applications represented and the hardware under study. This obscures any attempt to make a general rule formulation. Despite this difficulty, our paper concludes with recommendations for future exercises aimed at addressing these challenges.

## 2 Data collection

The collection process was coordinated and supervised to get the metric results, including meetings, reporting, and surveys conducted at different stages of the CMIP6 simulations (before, during, and after the simulation runs). All the partners listed in Table 2 were invited to participate in the tracking process. The coordination, meetings, and reporting were useful to evaluate the state of the collection from the partners, and we provided support to those institutions that required it during the collection process.

The data collection was divided into two steps: the initial phase comprehends the collection up to March 2020, coinciding with the first IS-ENES3 general assembly where the first results were presented; the second includes the data collected up to the end of 2020 when all the institutions had finished the CMIP6 runs. Finally, IS-ENES3 completed the last update to the Earth System Documentation [3] (ES-DOC) in the middle of 2021, publishing CPMIP along with the other CMIP6 results.

As the reader can see, not all institutions managed to provide the full set of CPMIP performance metrics. The metrics frequently missing are the *Coupling Cost*, *Memory Bloat*, and *Data Output cost*. This is primarily attributed to the challenges involved in their collection compared to metrics like *SYPD* or Parallelisation, which are well-known within the community and

---

[3]https://es-doc.org/

**Table 2.** List of institutions and models that provided the metrics from their CMIP6 executions. Also listed are the HPC platform and resolution used for the atmosphere (ATM) and ocean (OCN) components. Note that "resol" in Table 1 is defined as the number of gridpoints. For better readability, we present here this information using more conventional measure of degrees of latitude and longitude.

| Institution | Experiment | HPC machine | Atmosphere resol | Ocean resol | Reference |
|---|---|---|---|---|---|
| BSC | EC-Earth3 | MareNostrum4 | 0.7 | 1.0 | Döscher et al. (2022) |
| | EC-EarthVeg | | 0.7 | 1.0 | |
| CMCC | CM2-SR5 | Zeus | 1.0 | 1.0 | Lovato et al. (2022) |
| CNRM-CERFACS | CNRM-CM6-1-atm | Beaufix2 | 1.4 | | Voldoire et al. (2019) |
| | CNRM-CM6-1 | | 1.4 | 1.0 | |
| | CNRM-CM6-1-HR-atm | | 0.5 | | |
| | CNRM-CM6-1-HR | | 0.5 | 0.25 | |
| | CNRM-ESM2-1-atm | | 1.4 | | Séférian et al. (2019) |
| | CNRM-ESM2-1 | | 1.4 | 1.0 | |
| DKRZ | MPI-ESM1-HR | Mistral | 1.0 | 0.4 | Müller et al. (2018) |
| GFDL | OM4-p5 | Gaea | | 0.5 | Dunne et al. (2020) |
| | ESM4-piC | | 1.0 | 0.5 | |
| | CM4-piC | | 1.0 | 0.25 | |
| | OM4-p25 | | | 0.25 | |
| IITM | IITM-ESM | Intel AADITYA | 1.875 | 1.0 | Krishnan et al. (2021) |
| IMPE | BESM | xc50 | 1.875 | 1.0 | Veiga et al. (2019) |
| IPSL | IPSL-CM6A | Irene-SKL/Curie | 2.5 | 1.0 | Boucher et al. (2020) |
| KNMI | EC-Earth3 | Rhino | 0.7 | 1.0 | Döscher et al. (2022) |
| | EC-Earth3-AerChem | | 0.7 | 1.0 | |
| MPI | MPI-ESM1-LR-ATM | Mistral | 4.0 | | Müller et al. (2018) |
| | MPI-ESM1-LR-LAND | | | | |
| | MPI-ESM1-LR | | 1.875 | 1.5 | |
| NERC | UKESM1-AMIP | Archer xc30 | 4.0 | | Sellar et al. (2020) |
| | UKESM1-0-LL | | 1.875 | 1.0 | |
| | HadGEM3-GC3.1-LL | | 1.875 | 1.0 | |
| | HadGEM3-GC3.1-HM | | 0.8 | 0.25 | Williams et al. (2018), |
| | HadGEM3-GC3.1-HH | | 0.8 | 0.08 | |
| NorESM2 | NorESM2-LM | Fram | 2.5 | 1.0 | Seland et al. (2020) |
| | NorESM2-MM | | 1.0 | 1.0 | |
| SMHI | EC-EarthVeg | Tetralith/Beskow | 0.7 | 1.0 | Döscher et al. (2022) |
| UKMO | UKESM1-0-LL | xce xc40 | 1.875 | 1.0 | Sellar et al. (2020) |
| | HadGEM3-GC3.1-LL | | 1.875 | 1.0 | Williams et al. (2018) |
| | HadGEM3-GC3.1-MM | | 0.8 | 0.25 | |

relatively easier to obtain. Other impediments to collect the CPMIP metrics include time and resources constraints, particularly considering that the focus of the simulations are leans more towards science aspects than to the computational realm during
CMIP6 runs. Additionally, some institutions reported that changes in the underlying computational infrastructure have made the collection process more difficult.

**Table 3.** List of institutions with the model and CPMIP metrics. We also include the Useful Simulated Years (Useful SY), which accounts for the number of years simulated by each experiment that generated data with scientific value

| Institution | Experiment | Resol | Cmplx | SYPD | ASYPD | CHSY | Paral | JPSY | Cpl C | Mem B | DO | DI | Useful SY |
|---|---|---|---|---|---|---|---|---|---|---|---|---|---|
| BSC | EC-Earth3 | 1.99E+07 | 34 | 15.20 | 9.87 | 1213 | 768 | 4.41E+07 | 0.080 | 59.5 | 1.12 | 0.041 | 14020 |
| | EC-EarthVeg | 1.99E+07 | | 12.36 | 7.42 | 1491 | 768 | 4.87E+07 | 0.100 | 68.48 | 1.13 | 0.059 | 252 |
| CMCC | CM2-SR5 | 6.94E+06 | 397 | 6.68 | 6.50 | 2069 | 576 | 1.67E+09 | 0.074 | 17.8 | 1.04 | 0.050 | 965 |
| CNRM-CERFACS | CNRM-CM6-1-atm | 2.98E+06 | 128 | 7.30 | 6.10 | 1292 | 393 | 3.50E+07 | | | | | 5723 |
| | CNRM-CM6-1 | 1.02E+07 | 181 | 8.10 | 7.30 | 1352 | 400 | 3.38E+07 | | | | | 22241 |
| | CNRM-CM6-1-HR-atm | 2.36E+07 | 128 | 2.20 | 1.80 | 1541 | 520 | 4.80E+07 | | | | | 1190 |
| | CNRM-CM6-1-HR | 1.37E+08 | 181 | 1.50 | 1.48 | 4289 | 840 | 1.07E+08 | | | | | 1642 |
| | ESM2-1-atm | 2.98E+06 | 335 | 7.10 | 6.40 | 8520 | 781 | 2.28E+08 | | | | | 1759 |
| | ESM2-1 | 1.10E+07 | 393 | 4.70 | 4.40 | 21552 | 1347 | 5.28E+08 | | | | | 11761 |
| DKRZ | MPI-ESM1-HR | 2.00E+07 | | 13.33 | 11.00 | 4710 | 2616 | 3.21E+08 | | | | | 1864 |
| GFDL | OM4-p5 | 3.32E+07 | 13 | 15.90 | 12.22 | 1962 | 1300 | 7.50E+07 | 0.140 | 33.61 | | 0.039 | 300 |
| | ESM4-piC | 3.76E+07 | 140 | 8.65 | 7.46 | 13576 | 4893 | 5.19E+08 | 0.270 | 40.57 | | 0.003 | 1124 |
| | CM4-piC | 1.28E+08 | 31 | 9.98 | 8.16 | 15388 | 6399 | 3.72E+08 | 0.130 | 47.64 | | 0.018 | 657 |
| | OM4-p25 | 1.26E+08 | 11 | 11.50 | 7.05 | 9748 | 4671 | 5.88E+08 | 0.260 | 16.09 | | 0.006 | 300 |
| IITM | IESM | 1.83E+06 | 168 | 8.00 | 7.00 | 996 | 332 | 3.81E+07 | | 36.7 | | | 845 |
| IMPE | BESM | 6.88E+06 | 132 | 3.60 | 3.40 | 1853 | 278 | | | | | 0.020 | 360 |
| IPSL | IPSL-CM6A | 1.06E+07 | 750 | 12.00 | 11.50 | 1900 | 950 | 1.16E+08 | 0.050 | 10.00 | 1.20 | 0.070 | 53000 |
| KNMI | EC-Earth3 | 1.99E+07 | 34 | 16.20 | 16.20 | 1286 | 868 | | | | | | 1009 |
| | EC-Earth3-AerChem | 2.06E+07 | | 3.03 | 3.03 | 3549 | 448 | | | | | | 730 |
| MPI | MPI-ESM1-LR-ATM | 8.66E+05 | | 45.90 | 25.20 | 163 | 312 | 1.11E+07 | | | | | 991 |
| | MPI-ESM1-LR-LAND | 8.33E+05 | | 282.80 | 265.40 | 3 | 36 | 1.39E+06 | | | | | 2460 |
| | MPI-ESM1-LR | 3.12E+06 | | 55.60 | 22.70 | 379 | 878 | 2.56E+07 | | | | | 18860 |
| NERC | UKESM1-AMIP | 2.35E+06 | 202 | 1.64 | 1.41 | 7376 | 504 | 1.04E+08 | | 52.50 | 1.31 | 0.003 | 45 |
| | UKESM1-0-LL | 1.14E+07 | 252 | 2.02 | 1.10 | 8554 | 720 | 3.18E+08 | 0.078 | 28.00 | 1.19 | 0.005 | 195 |
| | HadGEM3-GC3.1-LL | 1.14E+07 | 150 | 4.25 | 1.06 | 12198 | 2160 | 4.33E+08 | 0.047 | 56.80 | 1.41 | 0.016 | 70 |
| | HadGEM3-GC3.1-HM | 1.99E+08 | 54 | 0.58 | 0.46 | 192662 | 4656 | 7.70E+09 | 0.210 | 154.00 | | 0.001 | 65 |
| | HadGEM3-GC3.1-HH | 1.26E+09 | 54 | 0.49 | 0.34 | 588931 | 12024 | 2.30E+10 | | 183.00 | 1.41 | 0.0004 | 65 |
| NorESM | NorESM2-LM | 1.01E+07 | | 13.84 | 3.03 | 1665 | 960 | 5.60E+07 | 0.035 | | | 0.065 | 5463 |
| | NorESM2-MM | 1.14E+07 | | 8.96 | 6.14 | 4886 | 1824 | 1.65E+08 | 0.32 | | | 0.060 | 1021 |
| SMHI | EC-EarthVeg | 1.99E+07 | | 12.44 | 6.65 | 1667 | 864 | | | | | 0.028 | 6337 |
| UKMO | HadGEM3-GC3.1-LL | 1.14E+07 | 228 | 4.00 | 3.55 | 13392 | 2232 | 4.97E+08 | 0.061 | 46.00 | 1.03 | 0.074 | 5610 |
| | UKESM1-0-LL | 1.14E+07 | 372 | 4.30 | 3.60 | 16074 | 2880 | 5.97E+08 | 0.098 | 4.60 | 1.03 | 0.019 | 15435 |
| | HadGEM3-GC3.1-MM | 1.44E+08 | 236 | 1.65 | 1.32 | 62836 | 4320 | 2.33E+09 | 0.105 | 120.00 | 1.02 | 0.050 | 2386 |

**Table 4.** List of HPC machines used for the experiments under study, detailing hardware specifications, benchmark results (Linpack and High-Performance Conjugate Gradient, HPCG), theoretical performance (Rpeak), power consumption and Power Usage Effectiveness (PUE) for each system

| Institution | Machine | total cores | cores per node | Mem node (GB) | Mem per core (GB) | network | CPU family | CPU freq (GHz) | Rpeak (PFlop/s) | Linpack (PFlop/s) | Power (kW) | HPCG (TFlop/s) | PUE |
|---|---|---|---|---|---|---|---|---|---|---|---|---|---|
| BSC | MN4 | 155520 | 48 | 96 | 2.00 | Intel Omni-Path | Platinum Skylake | 2.10 | 10.300 | 6.22 | 1632 | 122.24 | 1.35 |
| CMCC | Zeus | 12528 | 36 | 96 | 2.67 | InfiniBand | Gold Skylake | 3.00 | 1.202 | | | | 1.84 |
| CNRM-CERFACS | Beaufix2 | 73440 | 40 | 64 | 1.60 | InfiniBand | E5 Broadwell | 2.20 | 2.590 | 2.16 | 830 | 35.34 | |
| DKRZ/MPI | Mistral | 100200 | 30 | 68 | 2.25 | InfiniBand | E5 Haswell | 2.29 | 3.960 | 3.01 | 1116 | 44.11 | 1.19 |
| IITM | AADITYA | 38144 | 16 | 64 | 4.00 | InfiniBand | E5 Haswell | 2.60 | 0.790 | 0.72 | 790 | | |
| INPE | xc50 | 4080 | 40 | 192 | 4.80 | Aries Interconnect | Gold Skylake | 2.40 | 0.313 | | | | |
| IPSL | Curie | 80640 | 16 | 64 | 4.00 | InfiniBand | E5 Sandy Bridge | 2.70 | 1.670 | 1.36 | 2132 | 50.99 | 1.43 |
| IPSL | Irene | 79488 | 48 | 192 | 4.00 | InfiniBand | Platinum Skylake | 2.70 | 6.640 | 4.07 | 917 | 52.68 | |
| KNMI | Rhino | 4752 | 28 | 128 | 4.57 | InfiniBand | Nehalem | 3.06 | 0.058 | | | | |
| NERC | Archer xc30 | 118080 | 24 | 64 | 2.67 | Aries Interconnect | E5 Ivy | 2.70 | 2.550 | 1.64 | | 80.79 | 1.10 |
| NorESM | Fram | 32256 | 32 | 64 | 2.00 | InfiniBand | E5 Broadwell | 2.10 | 1.100 | 0.95 | | | |
| SMHI | Beskow | 65920 | 32 | 64 | 2.00 | Aries Interconnect | E5 Haswell | 2.30 | 2.440 | 1.80 | 842 | | |
| SMHI | Tetralith | 61056 | 32 | 96 | 3.00 | Intel Omni-Path | Gold Skylake | 2.10 | 4.340 | 2.97 | | 65.24 | |
| UKMO | xc40 | 241920 | 36 | 192 | 5.33 | Aries Interconnect | E5 Broadwell | 2.10 | 8.130 | 7.04 | | | 1.35 |

## 2.1 Additional data collected

The CPMIP metrics serve not only as a means of computational evaluation but also provide valuable insights for broader analysis. In light of this, we collaborated with the Carbon Footprint Group created within the IS-ENES3 consortium, which was responsible for evaluating the Total Energy Cost associated with the CMIP6 experiments

$$Total\ Energy\ Cost = Useful\ Simulated\ Years \times JPSY \tag{1}$$

The Total Energy Cost of an experiment is defined as the product of the Useful Simulated Years, defined as years of simulation that produced data with a scientific value that was either shared between the groups or kept within the producer group for scientific analysis, and the Jules per Simulated Year (JPSY). This collaboration enabled us to provide for the first time an estimation of the carbon footprint related to those experiments. The Carbon Footprint was calculated following Equation 2.

$$Carbon\ Footprint = Total\ Energy\ Cost \times CF \times PUE \tag{2}$$

where the *Total Energy Cost* is in MWh, *CF* is the greenhouse gas conversion factor from MWh to CO2 kilogram according to the supplier bill or the country energy mix, and *PUE* (*Power Usage Effectiveness*) accounts for other costs sustained from the data-center, such as cooling. The results for all the institutions that participated in the study during the CPMIP collection are shown during the analysis section in Table 10.

## 2.2 Uncertainty in the measurements

Understanding measurement uncertainty and machine variability has a significant role in any performance analysis, particularly when comparing models running across different platforms without advanced performance tools or methods like tracing or sampling. Before starting the collection of the metrics, we asked each institution to indicate the machine variability, which was reported to be below 10% for all machines used. This provides an initial rough estimation, subject to future refinement efforts like the usage of benchmarking codes for climate science like the one proposed by van Werkhoven et al. (2023).

It is important to note that not all metrics exhibit the same variability range. Certain metrics, such as Parallelisation, Resolution, platform, and model Complexity, are constant values determined just by the experimental configurations, the HPC infrastructure, and model characteristics. These are considered *static* metrics.

The rest of the metrics are related to the execution speed and therefore subject to different sources of variability. On the one hand metrics like the SYPD or CHSY are well-known by the community and straightforward to collect: this results in less margin of error during collection and any variability should be attributed solely to the machine. On the other hand metrics like the Actual SYPD, JPSY, Coupling Cost, Memory Bloat, Data Intensity, and Data Output Cost are less common to collect and this can lead to confusion and human errors (e.g. whether the Actual SYPD should include system interruptions or only queue time can bring a to systematic misreporting). This represents a second source of variability, difficult to assess and estimate.

Identifying and understanding this uncertainty is key for accurately interpreting and comparing the performance of models across different centers. Special effort has been made to ensure the quality and correctness of the metrics presented in this

work through continuous support of the groups during the collection and double-checking of the reported numbers with the responsible for each institution whenever needed.

Future collections like this one will contribute to better identify and address metrics uncertainty, while detailed analysis of individual metrics will enhance our understanding of their characteristics and exhibit variability. For instance, studies like Acosta et al. (2023) focus mainly on the Coupling Cost and offer valuable lessons for understanding and measuring this metric. Therefore, mitigating possible uncertainties arising from misconception or lack of the appropriate tools to collect them in the future.

## 3 Analysis

Analysing metrics derived from diverse models, executed on multiple platforms, and managed by independent institutions presents a non-trivial challenge. Moreover, the presence of missing values further complicates the analysis, making it difficult to substitute them with estimations or interpolations, particularly given the relatively limited size of the dataset.

Our approach consisted of first: validating the metrics provided by the institutions. We have sometimes found that the metrics reported for some models were orders of magnitude apart from the rest. In this case, we started actively communicating with the institutions asking them to double-check the values and assisting them in the re-computation process. After going through this process for each one of the metrics and models we came up with the values reported in Section 2: in Table 2 and Table 3 the reader can find the complete list of models for which the CPMIP metrics were collected, with the name of the institution that was in charge for the run, the resolution used for the OCN and ATM, the reference for the experiment configuration, and the CPMIP metrics. Additionally, we include in Table 4 the most relevant information on the HPC platforms used by the institutions and some supplementary metrics in Table 10 related to the execution costs in CO2 emissions.

Later, and for each of the metrics analysed in detail in the following sections, we filtered by model selecting those where the metric was provided, sorting and/or grouping them by the reported value. Finally, to uncover possible relations among the metrics, we have used both statistical approaches (e.g. Pearson's correlation, Freedman et al., 2007) and qualitative analysis.

## 3.1 Resolution

The first attempt to extract valuable information from Table 3 was done by grouping the experiments by resolution, since for the moment we want to compare the performance achieved by ESMs whose target is similar. We are ignoring here the fact that for some simulations the set-up has fewer grid points (e.g. reduced Gaussian in the atmosphere or removal of land points in the ocean) and we are using the total size of the corresponding regular grid. The resolution of a component is measured as the number of grid points it has (NX x NY x NZ), and the total resolution is given by the sum of the resolutions of their constituents. There is not a strict consensus on the connection between the number of grid points and the categorisation of low, medium, and high resolutions. Thus, for the grouping we have used both the naming provided by the institution in charge of the experiment and the total number of grid points used for each model configuration. Most configurations have been categorised as low resolution and use up to 2.10E+07 grid-points in total, or no less than 0.7 degrees latitude-longitude grid spacing for any

of the components (see Figure 1 and Table 2). On the other hand, only those experiments with an Ocean/Atmosphere resolution
under 0.5 degree are treated as medium-high resolution configurations (see Figure 2).

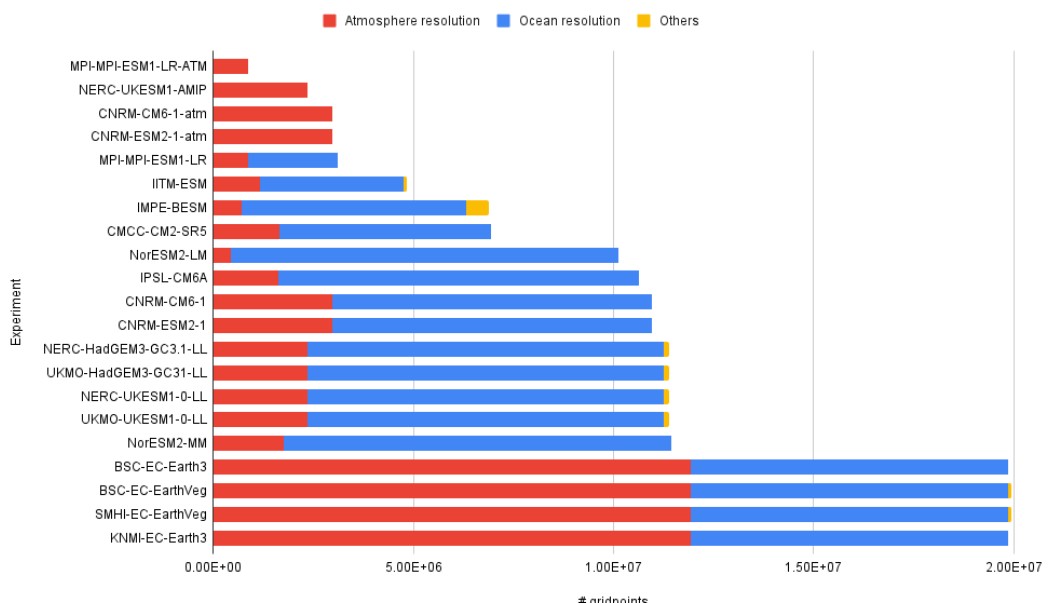

**Figure 1.** Atmosphere and ocean grid-points for low-resolution experiments. The yellow color refers to components that are contributing to the atmosphere or the ocean but can not be counted as a general circulation model per se (e.g. land-surface, sea ice, vegetation, etc.).

We see in Figure 1 the low-resolution experiments. The number of grid points for the ATM (red) and OCN (blue) components for each model/institution has been listed in ascending order. We observe that except for EC-Earth, all other models run the OCN at a higher resolution than the ATM. More precisely, the OCN resolution is between 3 to 5 times bigger for MPI-ESM,

BESM, CM2-SR2, CNRM-CM6, HadGEM3-LL, UKESM-LL and NorESM-MM. While in EC-Earth, it only accounts for 1/3 of the total model resolution (the remaining 2/3 are used for the ATM). Remarkably, the LM configuration used at NorESM uses a grid for the OCN which is 22 times bigger than the one for the ATM. As one would expect, the total number of grid points of an experiment can be explained solely by the ATM and OCN resolution used, but we will show later how adding more components/features (in yellow in Figure 1) can have quite an impact on the performance anyway.

Figure 2 shows the number of ocean and atmosphere grid points for the medium-high resolution experiments. We observe that like most of the low-resolution ones, all experiments use more grid points for the oceanic component than for the atmospheric one (notably, GFDL CM4-piC experiment use 55x more grid points in the OCN component). The ATM resolutions range between 1 and 0.4 degree, while OCN ones mostly run at 1/4 of a degree, except for the NERC-HadGEM3-GC3.1-HH experiment which runs the oceanic component at 1/12.

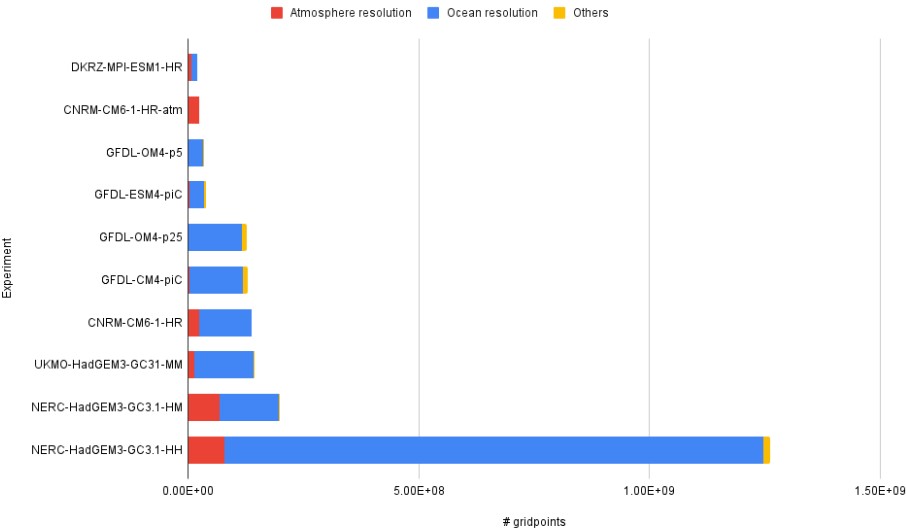

**Figure 2.** Atmosphere and ocean grid-points for medium-high resolution experiments

## 3.2 Complexity

The complexity of a coupled model, as defined in Table 1, is the number of prognostic variables among all components. Here, "prognostic" refers to variables that the model directly predicts, such as temperature, atmospheric humidity, salinity, etc. In other words, variables that can be obtained directly as outcomes of the model. This metric is not well-known by the community and never collected before, leading to confusion in some cases. Therefore, the values reported have are to be seen as approximations. Only by continuously measuring these metrics in future collections will we improve our understanding of model complexity and its implications on model performance. The data in Table 5 reveals a wide variability in *Complexity* (*Cmplx*) across the models, with most models reporting a value that ranges between 100 to 400. Notably, GFDL (OM4 and CM4) and EC-Earth have considerably lower *Cmplx*. IPSL-CM6A model stands out in this context with a *Cmplx* of 750, which is markedly higher than the other models, potentially due to its representation of the carbon cycle. Likewise, we were expecting a much higher value for the EC-Earth-Veg experiment, but it was impossible to get this metric for the vegetation component (LPJ-Guess) even after contacting the developers. This highlights the challenge of obtaining this metric with accuracy, partly due to a lack of awareness of the number of prognostic variables of the components among users of the ESMs, leading to an overestimation for this metric, and also because the approximation based on the size of the restart files (Balaji et al., 2017, p. 25) is not always accurate. For instance, LPJ-Guess restart file size can measure tens of GB and depends on the Parallelisation used for this component. What's more, explaining why NERC HadGEM3-GC31 *Cmplx* is almost 3 times larger for the lower resolution configuration (LL) than for the same experiment using more grid points (MM, HM and HH configurations) represents

**Table 5.** Resolution, SYPD, CHSY, Parallelisation and Coupling Cost for experiments that reported the Complexity metric

| Institution | Experiment | Resolution | SYPD | CHSY | Parallelisation | Complexity | Coupling Cost |
|---|---|---|---|---|---|---|---|
| BSC | EC-Earth3 | 1.99E+07 | 15.20 | 1491 | 768 | 34 | 0.080 |
| CNRM-CERFACS | CNRM-CM6-1-atm | 2.98E+06 | 7.30 | 1292 | 393 | 128 | |
| | CNRM-CM6-1 | 1.10E+07 | 8.10 | 1541 | 520 | 181 | |
| | CNRM-CM6-1-HR-atm | 2.36E+07 | 2.20 | 8520 | 781 | 128 | |
| | CNRM-CM6-1-HR | 1.37E+08 | 1.50 | 21552 | 1347 | 181 | |
| | ESM2-1-atm | 2.98E+06 | 7.10 | 1352 | 400 | 335 | |
| | ESM2-1 | 1.10E+07 | 4.70 | 4289 | 840 | 393 | |
| GFDL | OM4-p25 | 1.26E+08 | 11.50 | 9748 | 4671 | 11 | 0.130 |
| | OM4-p5 | 3.32E+07 | 15.90 | 1962 | 1300 | 13 | 0.140 |
| | CM4 | 1.28E+08 | 9.98 | 15388 | 6399 | 31 | 0.260 |
| | ESM4 | 3.76E+07 | 8.65 | 13576 | 4893 | 140 | 0.270 |
| IITM | IESM | 1.83E+06 | 8.00 | 996 | 332 | 168 | |
| IMPE | BESM | 6.88E+06 | 3.60 | 1853 | 278 | 132 | |
| IPSL | IPSL-CM6A | 1.06E+07 | 12.00 | 1900 | 950 | 750 | 0.050 |
| KNMI | EC-Earth3 | 1.99E+07 | 16.20 | 1286 | 868 | 34 | |
| NERC | HadGEM3-GC3.1-HM | 1.99E+08 | 0.58 | 192662 | 4656 | 54 | 0.210 |
| | HadGEM3-GC3.1-HH | 1.26E+09 | 0.49 | 588931 | 12024 | 54 | |
| | HadGEM3-GC3.1-LL | 1.14E+07 | 4.25 | 12198 | 2160 | 150 | 0.047 |
| | UKESM1-AMIP | 2.35E+06 | 1.64 | 7376 | 504 | 202 | |
| | UKESM1-0-LL | 1.14E+07 | 2.02 | 8554 | 720 | 252 | 0.078 |
| UKMO | HadGEM3-GC31-LL | 1.14E+07 | 4.00 | 13392 | 2232 | 228 | 0.061 |
| | HadGEM3-GC31-MM | 1.44E+08 | 1.65 | 62836 | 4320 | 236 | 0.105 |
| | UKESM1-0-LL | 1.14E+07 | 4.30 | 16074 | 2880 | 372 | 0.098 |

a challenge. Similarly, the notable differences between NERC and UMKO measurements, despite both running HadGEM-GC3.1 and UKESM1 models but on different platforms, raise questions about their source, which requires further investigation.

Nonetheless, the data from CNRM-CERFACS provides evidence supporting the idea that the *Cmplx* of a model should remain consistent regardless of the resolution, and only increase as additional features are simulated by the ESM. For instance, the *Cmplx* of CNRM-CM6 ATM standalone runs (CNRM-CM6-1-atm and CNRM-CM6-1-HR-atm) is 128 and grows up to 181 when the OCN component is included for the coupled configuration (CNRM-CM6-1 and CNRM-CM6-1-HR). The same is also observed for the CNRM-ESM2 model, where the *Cmplx* increases from 335 to 393 when adding the OCN component. Furthermore, in both cases, the ESMs require more processing elements when running the coupled version. This shows a clear interconnection between the Parallelisation and *Cmplx* as both will grow when comparing standalone and coupled simulations, other examples are: NERC standalone execution UKESM1-AMIP and UKESM1-LL coupled version, GFDL standalone OM4 (OCN only) runs and the coupled configurations ESM4 and CM4, and CNRM-CM6-atm (ATM only), CNRM-CM6-1 (ATM and OCN) and IPSL-CM6A (ATM, OCN and chemistry).

Therefore, *Cmplx* usually reduces the *SYPD* achieved and/or increases the *CHSY* given that adding a new component will, at best, only increase the latter. Maintaining the same throughput when increasing the *Cmplx* requires the use of more parallel resources, which translates into more costly executions and is usually correlated to parallel efficiency loss due to the need for

**Table 6.** Experiments that reported the Data Output cost (DO) and Data Intensity (DI) metrics

| Institution | Experiment | Resolution | Complexity | SYPD | CHSY | Parallelisation | DO | DI |
|---|---|---|---|---|---|---|---|---|
| BSC | EC-Earth3 | 1.99E+07 | 34 | 15.20 | 1213 | 768 | 1.12 | 0.0410 |
| | EC-EarthVeg | 1.99E+07 | | 12.36 | 1491 | 768 | 1.13 | 0.0590 |
| CMCC | CM2-SR5 | 6.94E+06 | 397 | 6.68 | 2069 | 576 | 1.04 | 0.0500 |
| GFDL | OM4-p5 | 3.32E+07 | 13 | 15.90 | 1962 | 1300 | | 0.0392 |
| | OM4-p25 | 1.26E+08 | 11 | 11.50 | 9748 | 4671 | | 0.0178 |
| | ESM4-piC | 3.76E+07 | 140 | 8.65 | 13576 | 4893 | | 0.0032 |
| | CM4-piC | 1.28E+08 | 31 | 9.98 | 15388 | 6399 | 1.24 | 0.0058 |
| IMPE | IMPE-BESM | 6.88E+06 | 132 | 3.60 | 1853 | 278 | | 0.0200 |
| IPSL | IPSL-CM6A | 1.06E+07 | 750 | 12.00 | 1900 | 950 | 1.20 | 0.0700 |
| NERC | HadGEM3-GC3.1-LL | 1.14E+07 | 150 | 4.25 | 12198 | 2160 | 1.41 | 0.0160 |
| | HadGEM3-GC3.1-HM | 1.99E+08 | 54 | 0.58 | 192662 | 4656 | | 0.0006 |
| | HadGEM3-GC3.1-HH | 1.26E+09 | 54 | 0.49 | 588931 | 12024 | 1.41 | 0.0004 |
| | UKESM1-AMIP | 2.35E+06 | 202 | 1.64 | 7376 | 504 | 1.31 | 0.0030 |
| | UKESM1-0-LL | 1.14E+07 | 252 | 2.02 | 8554 | 720 | 1.19 | 0.0050 |
| NorESM | NorESM2-LM | 1.01E+07 | | 13.84 | 1665 | 960 | | 0.0650 |
| | NorESM2-MM | 1.14E+07 | | 8.96 | 4886 | 1824 | | 0.0600 |
| SMHI | EC-EarthVeg | 1.99E+07 | | 12.44 | 1667 | 864 | | 0.0280 |
| UKMO | UKESM1-0-LL | 1.14E+07 | 372 | 4.30 | 16074 | 2880 | 1.03 | 0.0190 |
| | HadGEM3-GC31-LL | 1.14E+07 | 228 | 4.00 | 13392 | 2232 | 1.03 | 0.0740 |
| | HadGEM3-GC31-MM | 1.44E+08 | 236 | 1.65 | 62836 | 4320 | 1.02 | 0.0500 |

coupling synchronisations and interpolations (e.g. see GFDL results in Table 5). The relation between *Cmplx* and the Coupling Cost is further discussed in subsection 3.5.

## 3.3 Data output

ESMs generate a large amount of output data, including model results, diagnostics, and intermediate variables, which need to be written to storage. Writing and saving this massive amount of data to disk or other storage mediums is time-consuming and can affect the overall performance of the model. Concurrent access to storage resources by multiple processes or multiple model instances can create contention, may represent an I/O bottleneck, and eventually degrade performance and scalability. CPMIP metrics add two metrics to quantify and evaluate the I/O workload: the *Data Output Cost* (*DO*), which reflects the cost

of performing I/O and is determined as the ratio of *CHSY* with and without I/O; and the *Data Intensity* (*DI*), which measures the data production efficiency in terms of data generated per compute hour (i.e. GB/Core-hour).

Data Output Cost

From Table 6, we see that all the experiments conducted by UKMO and CMCC reported a *DO* below 1.05, even though the *DI* varies considerably between the different experiments. Moreover, we observe that the *DO* is much higher for the same

ESM (HadGEM-GC31-LL and UKESM1-0-LL) when executed by NERC, reaching 1.19 for UKESM1-0-LL and 1.41 for HadGEM3-GC31-LL. It is not possible to know, however, if this is due to the difference between the HPC platform used or to differences in the model I/O configuration. This underscores the importance of of the specific model's I/O configuration in

influencing the *DO* metric. Besides, neither the metrics collected from UKMO nor the ones reported from NERC show that the *DO* should increase when running higher-resolution experiments (HadGEM3-GC31-MM and HH configurations). Moreover, EC-Earth and EC-Earth-Veg *DO* measurements are almost the same, suggesting that adding the vegetation model to EC-Earth does not increase the cost of the I/O, while UKESM runs conducted by NERC show that the *DO* is much higher when running the ATM standalone configuration, UKESM-AMIP, that the coupled run, UKESM-1-LL. Thus, the increase in Complexity or Resolution does not increase the cost of the I/O but the cost of the whole ESM simulation, which can diminish the *DO* metric if I/O workload stays constant.

### Data Intensity

As seen in Table 6, the *DI* is generally of the order of MB per core-hour and gets smaller as we move to higher-resolution experiments (i.e. higher *CHSY*), meaning that the amount of data generated does not grow proportionally with the number of grid points nor with the execution cost. For instance, the *DI* reported for NERC-HadGEM, UKMO-HadGEM, NorESM2 and GFLD-OM4 experiments decreases when increasing the resolution. Thus, we observe a positive correlation between the *SYPD* and the *DI*.

## 3.4 Workflow and infrastructure costs

The real execution time of climate experiments can not be explained only by the speed at which a model can run. Queue times before having access to the HPC resources (usually managed by an external scheduler), service disruption, errors in the model/workflow manager, etc. can heavily extend the time-to-solution of ESMs. From the data in Table 1, we see that the difference between the *SYPD* and *ASYPD* reported varies a lot between institutions. Some claim that they had no overhead in their runs (KNMI), while for others it can account for up to 78% (NorESM2-LR). The histogram in Figure 3 helps illustrate the spread of the *ASYPD* overhead: it rarely surpasses 50% and half of the institutions reported it to be less than 20%. Judging from the spread of this metric and from the discussions after the collection, we consider that there are two groups: 1) Institutions that included solely the queue time, which reported an overhead under 20%, and 2) Institutions including not only the queue time but also the system interruptions and/or workflow management, which reported much higher values.

The results support the idea that queuing time represents an increment of around 10-20% to the speed of the ESM. On the other hand, adding interruptions and workflow management the total execution time could increase up to 40-50% compared to the simulation time alone. We do not have enough supporting data to draw any definitive conclusions, so we believe that it would be essential to add finer granularity to the *ASYPD* metric to be able to differentiate both factors. BSC CMIP6 results using the same configuration on two different platforms (Marenostrum and CCA) proved that the percentage of each part (queue time, interruptions or post-processing) could change among platforms even though the CMIP6 experiment is the same[4]. From the metrics listed in Table 3, we see that the difference between *SYPD* and *ASYPD* for the same model can significantly vary depending on the machine used for execution. For EC-Earth3 (standard and vegetation experiments), the overhead ranges from 0% at KNMI to 0.35-0.40% at BSC and up to 0.47% at SMHI. However, it is important to note that the value provided

---

[4]https://shorturl.at/lzAHO, retrieved February 6, 2024

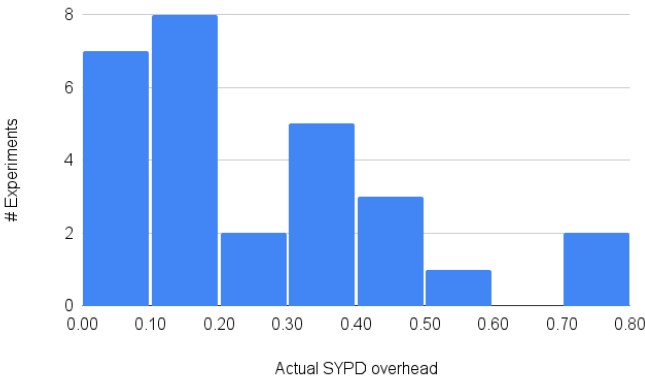

**Figure 3.** Histogram of the Actual SYPD overhead

by KNMI only accounts for the queue time, and they reported having instant access to the HPC resources. Furthermore, for HadGEM3-GC3.1-LL, we observe that NERC and UKMO runs are similar in the model execution speed, achieving approx. 4 *SYPD*, but totally different in the *ASYPD*. The overhead due to the workflow at UKMO is just 11%, whereas at NERC it takes 75%. We see something similar when comparing the same institutions for the UKESM-LL execution, where the overhead in UKMO is almost the same as before (16%), but it has drastically decreased at NERC. As we expected, the *ASYPD* overhead is related to the model *SYPD*, but more importantly to the workload of the platform used for the runs. Furthermore, we observed that for UKMO and MPI the smaller the Parallelisation, the smaller the overhead due to the workflow.

### 3.5 Coupling Cost

*Coupling Cost* (*Cpl C*, Equation 3) is an essential metric evaluated in this study. It quantifies the overhead introduced by coupling within an Earth System Model (ESM). This overhead encompasses various factors, including the coupling algorithms used for grid interpolations and calculations for conservative coupling. Additionally, it incorporates the impact of the load imbalance, which arises when different independent components of the ESM finish their computations at varying rates, potentially leaving processing elements idle. It is defined as follows:

$$Cpl\_Cost \equiv \frac{T_M P_M - \sum_c T_C P_C}{T_M P_M} \tag{3}$$

Where $T_M$ and $P_M$ are the runtime and Parallelisation for the whole coupled model, and $T_C$ and $P_C$ the same for each individual component it uses.

Figure 4 shows the list of institutions ordered from lower to higher *Cpl C*. Most institutions reported that the cost increase due to the coupling accounts for around 5-15% of the total. Only 4 (over the 16 that reported this metric) show an increase of over 20%. The data from GFDL (OM4-p5, OM4-p25, ESM4-piC, and CM4-piC) and UKMO (UKESM-LL and UKESM-AMIP) suggests that the increase in Complexity leads to higher *Cpl C* and lower *SYPD*. This aligns with the expectations, as the addition of a new component to the ESM will likely slow down the model and make the load balancing harder. It is

noteworthy that a similar trend is observed in EC-Earth experiments. Even though we don't know the exact value for EC-EarthVeg *Cmplx*, it is known to be higher than in the standard EC-Earth (ATM-OCN) configuration due to the inclusion of vegetation and chemistry models. When comparing the performance of these two runs, we see a decrease in the *SYPD* and a concurrent increase in the *Cmplx* and *Cpl. C*, as discussed in more detail in subsection 3.2.

In general, the *Cpl C* tends to rise when running experiments that use a higher Parallelisation. This could reflect a problem in the coupling phase. It can occur that the coupling algorithm is not scaling correctly or that the higher resolution configuration is not well-balanced. It is also likely that since the computing cost of running configurations in lower resolutions is smaller and less time-consuming, institutions can afford to run more spin-up tests and come up with a better distribution of processes among the coupled components to obtain a better load balance. In comparison, the contrary will happen for higher resolutions.

Since there are no specific tools to balance a coupled model, these institutions are forced to use a trial-and-error approach, which is not trivial for complex configurations with several components and/or differences in the time-stepping among them.

For these cases, a finer granularity in the *Cpl C* metric and new ways to achieve a well-balanced configuration could be needed, splitting interpolation algorithm and waiting time in different sub-metrics or providing some of the CPMIPs (*SYPD*, *CHSY . . .*) not only for the coupled version but also per component.

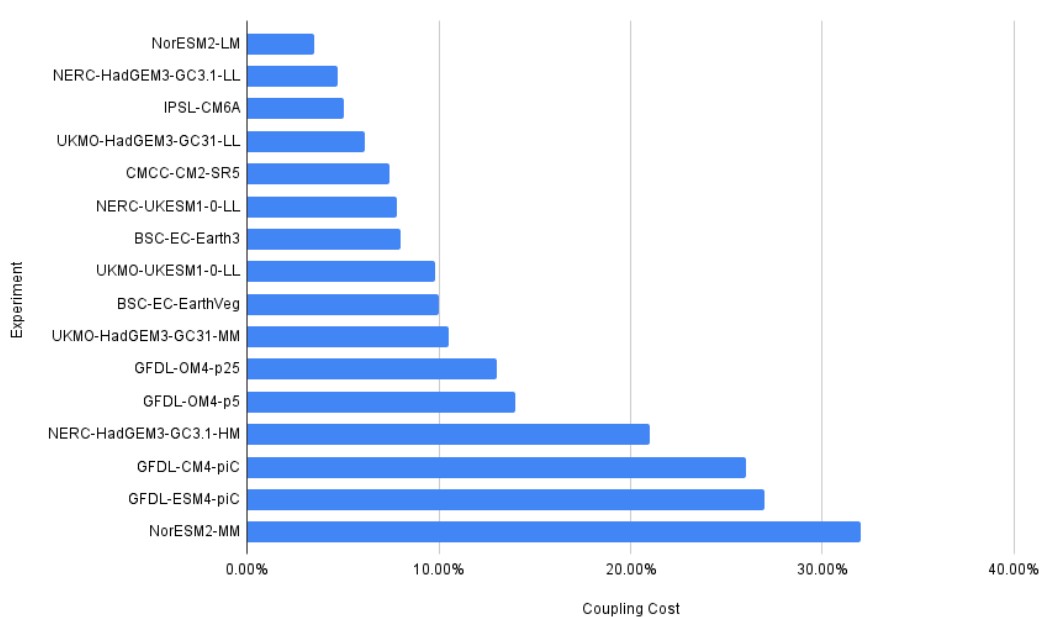

**Figure 4.** Coupling cost for all the institutions that provided the metric

**Table 7.** Metrics for models available on different HPC platforms

| Experiment | Institution | Resolution | SYPD | CHSY | Parallelisation |
|---|---|---|---|---|---|
| EC-Earth3 | BSC | 1.99E+07 | 15.20 | 1213 | 768 |
| | KNMI | 1.99E+07 | 16.20 | 1286 | 868 |
| EC-Earth3Veg | BSC | 1.99E+07 | 12.36 | 1491 | 768 |
| | SMHI | 1.99E+07 | 12.44 | 1667 | 864 |
| HadGEM3-GC3.1-LL | NERC | 1.14E+07 | 4.25 | 12198 | 2160 |
| | UKMO | 1.14E+07 | 4.00 | 13392 | 2232 |
| UKESM1-0-LL | NERC | 1.14E+07 | 2.02 | 8554 | 720 |
| | UKMO | 1.14E+07 | 4.30 | 16074 | 2880 |

## 3.6 Speed, cost, and Parallelisation

The *speed* of execution (*SYPD*) of a model is a fundamental metric that requires careful consideration. However, taken alone, it may not be enough to shed light on the model's performance itself. The meaning of a model's speed can only be fully understood when correlated to other important metrics. Among these the *Parallelisation* (*Paral*, i.e. the number of parallel resources allocated) stands out as a factor closely related to the speed and that, at the same time, directly influences the computational *cost* (*CHSY*) of the model execution. In this section, we show a detailed analysis of these three interconnected metrics. Contrary to what one would expect, the *SYPD* achieved by the models in this study is not always related to the resolution used nor to the *Paral* allocated. Although if we analyse how the same model performs on different HPC machines (Table 7), we note that higher values of *Paral* usually correspond to faster but more energy-consuming simulations.

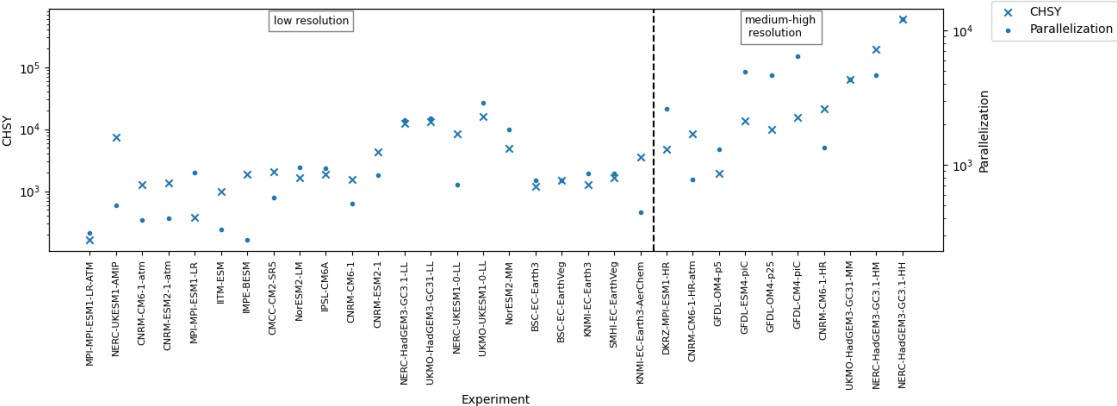

**Figure 5.** Comparison between CHSY and Parallelisation for both low and medium-high resolution experiments. Experiment configurations are arranged from left to right in ascending number of gridpoints. Note that vertical axis use a logarithmic scale for better visualisation

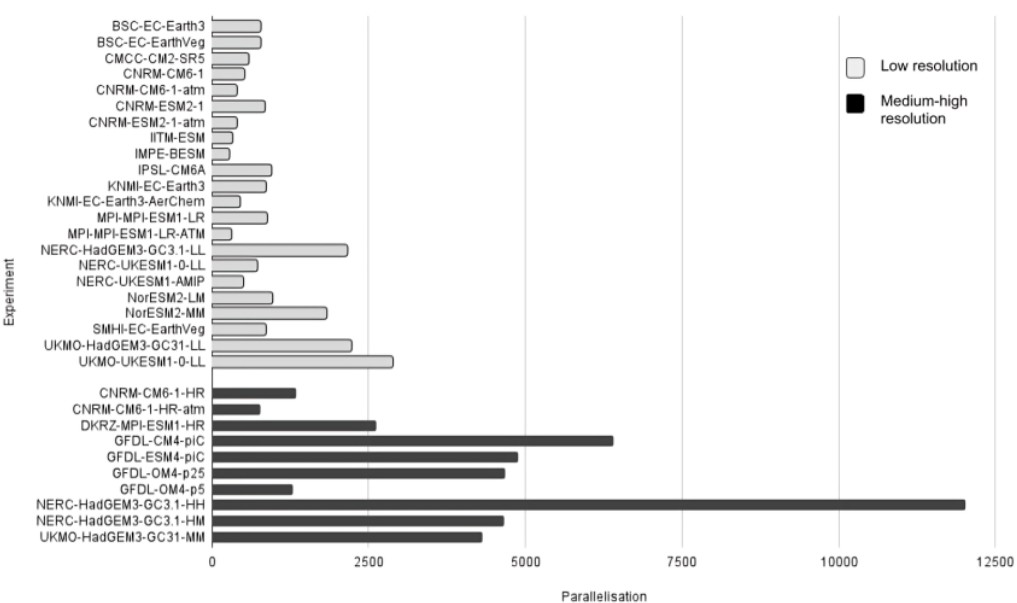

**Figure 6.** Parallelisation for low (grey) and medium-high (black) resolution models

As seen in Figure 5, the *Paral* and the *CHSY* are closely correlated in low-resolution models (e.g. CMCC-CM2-SR5,
NorESM2-LM, IPSL-CM6A, NERC-HadGEM3-GC3.1-LL, UKMO-HadGEM3-GC3.1-LL, UKMO-UKESM1-0-LL, NorESM2-
MM, BSC-EC-Earth3, BSC-EC-EarthVeg, KNMI-EC-Earth3, SMHI-EC-EarthVeg), showing that models do not scale in the
current generation of HPC platforms. Otherwise, one would see that the CHSY of ESMs with similar *Resol* do not increase
when using more processors given that the models run faster (i.e. higher *SYPD*). From the data, it is also clear which models are
under-performing. Take for instance KNMI-EC-Earth3-AerChem, despite using a smaller Parallelisation compared to its fam-
ily counterparts (BSC-EC-Earth3, BSC-EC-EarthVeg, KNMI-EC-Earth3 and SMHI-EC-EarthVeg), exhibits a higher CHSY.
Similarly, NERC-UKESM1-AMIP and NERC-UKESM1-0-LL employ less Parallelisation compared to UKMO-UKESM1-0-
LL, yet the CHSY does not decrease proportionally. Also, as illustrated in Figure 6, the level of Parallelisation tends to increase
as we move to higher-resolution experiments. Thus, and given that we do not observe a relation between the Resolution and the
*SYPD* achieved, we conclude that most institutions try to maintain at high-medium resolution the same *SYPD* achieved when
running lower-resolution configurations, at the cost of increasing the *CHSY*. Future collections that include more medium-high
resolution experiments will help creating further relationships for these experiments.

In addition, and already discussed in subsection 3.5, the *Cpl C* grows together with the Parallelisation, although there isn't
any sign that it limits the speed of the models.

**Table 8.** Resolution, SYPD, CHSY, Paral and Memory Bloat results for UKESM, EC-Earth and HadGEM3-GC31 experiments

| Experiment | Resolution | SYPD | CHSY | Parallelisation | Memory Bloat |
|---|---|---|---|---|---|
| BSC-EC-Earth3 | 1.99E+07 | 15.2 | 1213 | 768 | 59.50 |
| BSC-EC-EarthVeg | 1.99E+07 | 12.4 | 1491 | 768 | 68.48 |
| NERC-HadGEM3-GC3.1-LL | 1.14E+07 | 4.3 | 12198 | 2160 | 56.80 |
| NERC-HadGEM3-GC3.1-HM | 1.99E+08 | 0.6 | 192662 | 4656 | 154.00 |
| NERC-HadGEM3-GC3.1-HH | 1.26E+09 | 0.5 | 588931 | 12024 | 183.00 |
| UKMO-HadGEM3-GC31-LL | 1.14E+07 | 4.0 | 13392 | 2232 | 46.00 |
| UKMO-HadGEM3-GC31-MM | 1.44E+08 | 1.7 | 62836 | 4320 | 120.00 |
| NERC-UKESM-AMIP | 2.35E+06 | 1.6 | 7376 | 504 | 52.50 |
| NERC-UKESM-LL | 1.14E+07 | 2.0 | 8554 | 720 | 28.00 |

## 3.7 Memory Bloat

The *Memory Bloat* (Mem B, Equation 4) is the only CPMIP metric to evaluates models' memory usage by computing the ratio between the real to the ideal memory size. It is defined as:

$$Memory\_Bloat \equiv \frac{M - Parallelisation \cdot X}{M_i} \tag{4}$$

Where $M$ is the actual memory size, $X$ is the binary file size and $M_i$ the ideal memory size. The ideal memory size represents the size of the complete model state, which can be obtained by exploring the restart file size. This ratio is typically falls between

10-100. Large *Mem B* values signal of excessive buffering. As an example (Balaji et al. (2017)), for a rectangular grid with a halo size of 2 in X and Y directions, and a 20x20 domain decomposition, the 2-D array including halos is 576 (24x24) instead of 400 (20x20), resulting in a bloat factor of 1.44. Similarly, a 10x10 decomposition would yield an array area of 196, and a bloat ratio of 1.96.

   Table 8 presents the *Mem B* values reported for various models along other CPMIP metrics. We observe how the *Mem*

*B* increases with the resolution (e.g. NERC-HadGEM31), likely due to larger subdomains assigned to each compute unit in higher resolutions if the Parallelisation does not increase proportionally. Additionally, *Mem B* also increases when Complexity grows but the Parallelisation remains constant (e.g. BSC-EC-Earth3 with and without the vegetation model) as it requires keeping more data in memory. It is important to acknowledge the challenges in obtaining accurate memory usage for such applications, and the authors are aware that institutions faced difficulties in providing this data. Therefore, the reliability of the

reported values varies between sources, and should be contrasted by future measurements (e.g. CPMIP collection for CMIP7). Precise memory measurements, however, can only achievable with more advanced tools and approaches (memory profilers, MPI environment variables, etc.).

**Table 9.** Other CMIP6 measurements. The "Useful" metric, whenever used, accounts only for experiments that led to scientific value. The Power Usage Effectiveness (PUE) depends on the HPC machine used (Table 4)

| Institution | Useful Simulated Years* | Total Simulated Years | Useful Data Produced (PB) | Total Data Produced (PB) | Useful core hours (millions) | Total core hours (millions) | Total Person/Months | Total Energy Cost (TeraJoules) | PUE | Conversion Factor (MWh - kg CO2eq) | Carbon Footprint (tons CO2) |
|---|---|---|---|---|---|---|---|---|---|---|---|
| CMCC | 965 | | 0.097 | | 1.99 | | 7 | 1.61 | 1.84 | 408 | 329 |
| CNRM-CERFACS | 47,000 | | 1.350 | 2.48 | 160.00 | 365.00 | 450 | 6.18 | 1.43 | 40 | 97 |
| DKRZ | 1,276 | 1,321 | 0.600 | | 5.52 | 5.90 | | 0.41 | 1.19 | 184 | 24 |
| EC-Earth | 28,105 | 38,854 | 0.800 | 1.41 | 31.13 | 46.36 | 115 | 1.24 | 1.35 | 357 | 165 |
| IPSL | 75,000 | 165,000 | 1.800 | 7.60 | 150.00 | 320.00 | 200 | 8.72 | 1.43 | 50 | 172 |
| MPI-M | 24,175 | 35,000 | 1.930 | | 16.31 | | | 0.62 | 1.19 | 184 | 37 |
| NCC-NorESM2 | 23,096 | | 0.600 | | 27.23 | 80.00 | 150 | 1.69 | | | |
| NERC | 640 | | 0.460 | | 55.50 | | | 2.17 | 1.10 | 0 | 0 |
| UKMO | 37,237 | | 10.400 | | 683.00 | | | 26.70 | 1.35 | 87 | 868 |

. *The Useful Simulated Years column values can differ from Table 1 given that some of the experiment runs were not shown in that table

## 3.8 Carbon footprint

In addition to the CPMIP collection, we have also gathered the general metrics shown in Table 10. These metrics provide both useful (only accounting for simulations that produced data with scientific value) and total (encompassing all simulations, including spin-up and any runs that were finally discarded) numbers for the complete execution of CMIP6 experiments at the different institutions. They can be used to provide an idea about the total and useful number of years simulated, data produced and core hours consumed to finish the European community CMIP6 experiments. Although we did our best to collect the most updated data, we are aware that these numbers could have changed since the data collection was finished. We know that some institutions were doing some minor and final executions and updating databases such as ESGF. However, we consider Table 10 a very good representation of the effort done for the collection during CMIP6. In any case and taking into account the previous reasons, we do not analyse the results themselves and we will use this information to evaluate the Carbon Footprint associated with running models for large-scale projects like CMIP6, which is also a very interesting example for the community. By considering the useful Simulated Years, the HPC machine efficiency, and the KWH to CO2 conversion rates provided by each energy supplier, we calculated the Carbon Footprint (in tons of CO2) using Equation 2. As the reader can see, NERC reported a zero Carbon Footprint due to their green tariff power supplier. Among other institutions, CMCC is the one with the highest CF, followed by EC-Earth. Both significantly surpass the emissions of the other institutions: CERFACS, MPI, and UKMO have very small CO2 emissions per kWh. Regarding machine efficiency, CMCC reported that Zeus is the least power-efficient machine, with a Power Usage Effectiveness (PUE) of 1.84. CERFACS, IPSL, EC-Earth and UKMO reported similar values for their machines, while DKRZ, MPI-M and NERC have reported a PUE under 1.2. We believe that CMCC's Carbon Footprint may be overestimated, considering they simulated fewer than 1000 years yet reported nearly double the CO2 emissions compared to EC-Earth or IPSL, despite these institutions having simulated longer experiments (in SY). The Total Energy Cost of UKMO seems too big compared to their reported Useful Simulated Years. However, this can be attributed to the cost of maintaining the useful data produced, which amounts to 10.4 PB of disk space. The total Carbon Footprint is 1692 tCO2, even when

accounting for the experiments executed by only 8 out of the 49 institutions that are enlisted in CMIP6[5]. Based on a 2018 study by Acosta et al. Acosta and Bretonnière (2018), the Earth science group at the BSC, comprising around 80 people, had a CO2 equivalent of commuting (29 tCO2eq/yr), computing infrastructure (397 tCO2eq/yr), building and infrastructure (117 tCO2eq/yr), and travel (255 tCO2eq/yr). The total budget was, therefore, estimated to be near 800 tCO2eq/yr. Consequently, the carbon footprint from the execution of only this small subset of experiments more than doubles our budget in a single year.

This finding is consistent with observations from other groups within the community, such as a similar study conducted by CERFACS between 2019 and 2021, which reported a total budget of around 700 tCO2eq/yr. Nonetheless, the contributions that CMIP6 has to climate science are invaluable and beyond the immediate costs associated to running the simulations.

## 4   Drawbacks and actions recommended

Thanks to the experience learned from the data collection and analysis done, we recognise the importance of highlighting the
specific drawbacks we have found during this first collection as well as our recommendations to improve the collection and analysis for future iterations of multi-model climate research projects, such as CMIP7. The authors will continue working on this topic in the future not only to provide new approaches to facilitate the collection, but also in fostering the collaboration of the weather and climate science community to address the computational challenges of Earth modelling. Table 10 shows a list of the main drawbacks along suggested actions for improvement.

---

[5]https://wcrp-cmip.github.io/CMIP6_CVs/docs/CMIP6_institution_id.html

**Table 10.** Drawbacks and Recommended Actions for CMIP6 Metrics

| Drawbacks | Recommended Actions |
|---|---|
| CPMIPs are not enough to compare the performance of different ESMs running on different HPC platforms. | Multi-model comparisons will be better grounded once more data is available. Integrating the CPMIPs in the High-Performance Climate and Weather (HPCW, van Werkhoven et al., 2023) benchmark to evaluate the performance of the different machines used by the community. |
| Lack of resources and time to collect metrics after CMIP experiments. | Perform metric collection before or during CMIP experiments. Develop portable and automated processes for efficient collection. |
| Inconsistencies in metric collection hinder inter-model comparisons. | Normalise metric collection methods across institutions before multi-model runs. Develop tools to automatise the collection (e.g. integrated into the workflow manager). |
| Difficulty in identifying computational bottlenecks due to limited information. | Split sensitive metrics into sub-metrics for finer analysis. For instance, the Coupling Cost should separate interpolation from load-imbalance cost, and the ASYPD should differentiate between queue time and system interruptions. |

## 5  Conclusions

One of the limiting factors for climate science is the computational performance that Earth System Models (ESM) can achieve on modern High-Performance Computing (HPC) platforms. This limitation imposes constraints on the number of years that can be simulated, the number of ensembles that can be used, the resolution used by the models, the number of features simulated in one experiment, I/O intensity, data diagnostics calculated during the run, etc. Evaluating the performance of an ESM is a tremendous amount of work that generally requires: profiling the application, using tools to visualise and understand the profiling information, and developing and applying solutions based on the bottlenecks found. This process becomes even more complex when dealing with models used in large-scale multi-model projects like CMIP6, where multiple ESM are executed by different institutions that have access to diverse HPC platforms. To address these challenges, the Computational Performance Model Intercomparison Project (CPMIP) metrics were designed to be: universally available, easy to collect, and representative of the actual performance of ESMs and of the entire life-cycle of modeling (i.e. simulation and workflow costs).

This paper presents, for the first time, the results obtained from the CPMIP collection during the CMIP6 exercise. It provides the list of 14 institutions involved, primarily from the IS-ENES3 consortium, along with the 33 CMIP6 experiment configurations and the CPMIP metrics collected for each experiment. Furthermore, it goes well beyond mere data presentation and offers in-depth analysis for each metric collected to demonstrate the broader utility of the CPMIP collection. For instance, this

study investigates the resolution used by each model on the oceanic and atmospheric components, explores the relationship between execution speed and cost with the other metrics, assesses the impact of running models with higher processor counts, complexity, or I/O requirements, examines the overhead caused by queuing and workflow management, explores the coupling cost across different configurations, etc.

    Besides the CPMIP metrics analysis, this paper highlights results obtained from collaborations with other groups, such as

the Carbon Footprint Group. This collaboration underscore the shared concern of multiple institutions regarding computational performance in climate science and the joint effort to estimate the carbon footprint of the simulations conducted during the CMIP6 exercise.

    Finally, the paper addresses the main issues and drawbacks encountered during the collection and analysis of the metrics, including the heterogeneity of the models and HPC machines used, as well as uncertainty in the metric measurements reported.

These points should be of particular interest to the partners, aiming to improve and facilitate future collections. The paper also proposes recommendations to confront these challenges, which can be adopted by the community for the development of novel tools and more finely-grained metrics that would facilitate upcoming similar works. Moreover, the improvement and development of benchmarks specially designed for climate science will significantly enhance multi-platform performance comparisons. Continuous collection of these metrics in future multi-model projects (e.g. CMIP7) will facilitate the development

of a shared database for the community and vendors.

*Data availability.* The original data used during this work can be found here: http://bit.ly/3Y6XhHM

*Author contributions.* M.C. Acosta led the data collection process, ensuring that multiple institutions provided the necessary data and providing technical assistance throughout. S. Palomas conducted the data analysis and validation, and took the lead in writing the manuscript. S.P. Ticco contributed to data analysis and played a key role in revising and correcting the manuscript. Other co-authors were responsible for

data collection from their respective models and institutions

*Competing interests.* The contact author declares that Sophie Valcke, a co-author of this article, serves as a member of the editorial board of the Geoscientific Model Development (GMD) journal. The authors have no other competing interests to declare.

*Acknowledgements.* The research leading to these results has received funding from the EU H2020 ISENES3, under grant agreement n° 824084 and co-funding from the Spanish National Research Council through OEMES (PID2020-116324RA-I00)

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
