# Peer review of "The computational and energy cost of simulation and storage for climate science: lessons from CMIP6"

_Geoscientific Model Development, 2023_

## Author Comment (AC1)

Answer from authors to Reviewer 1 comments

**The computational and energy cost of simulation and storage for climate science: lessons from CMIP6**

**by**

Mario C. Acosta, Sergi Palomas, Stella Paronuzzi and Gladys Utrera

We sincerely appreciate the reviewer's insights, which have contributed to improving the clarity of certain sections of the manuscript, refining key concept definitions, and to better specify the context of the work. We are grateful for their valuable input, which has enhanced the overall quality of our work. Below, you will find our responses to the feedback provided in more detail:

**General comments**

This is an interesting and perhaps unique accounting of a subset of the models that participated in CMIP6 with a focus on model characteristics related to computing and carbon footprint details. My major comment is that the authors need to be clear that the models described in the paper are, in fact, a subset of the total number of models that participated in CMIP6. This subset is apparently the group of models that participated in the IS-ENES3 project. This needs to be made clear in the abstract and elsewhere.

**Detailed comments:**

List of authors: One of the coauthor's names is in the wrong order and misspelled: "Joussame Sylvie" should be "Sylvie Joussaume"

The error has been fixed in the manuscript, and we contacted the journal to update it as well.

Line 4: Here is an example of where the authors need to clarify the scope of the paper. I'd suggest the wording be changed to,

"This paper shows the main results obtained from the collection of performance metrics from 30 models that participated in the IS-ENES3 and represent a subset of the total of 124 CMIP6 models. The document provides…"

We have updated the introduction to present IS-ENES3 consortium and clarify that we only collected the metrics for a subset of CMIP6 experiments.

HOWEVER, it's unclear exactly how many models are actually involved.  I got the number 30 from line 42, but there are 32 models listed in Table 2, and 33 listed in Table 3.

The mismatch between the tables has been fixed. There are a total of 33 different experiments. We've substituted the term "model" with "experiment", which is more accurate and does not lead to confusion, given that a coupled experiment simulates multiple models.
We have modified the abstract, introduction and conclusions to state more clearly the institutions and experiments involved in the collection, and how many of them appear in the paper.

Line 41-42:  Once again, the authors need to be clear how their models relate to the larger CMIP6 set of models.  I recommend the following wording:  "In this paper, we present in Sec. 2 the collection of CPMIP metrics from the 30 [or 32, or 33] models that participated in the IS-NES3 project (Joussaume, 2010), out of the total of 124 CMIP6 models, and were used to simulate almost 500,000 years…"

We have checked and fixed mismatches previously appearing in the text, following the reviewer's advice.

Table 2 caption:  Pursuant to the comments above, the authors need to be clear how the models they list in this table relate to the total number of CMIP6 models.  This would avoid a scientist reading this paper and looking at this table and not seeing their model and wondering why they aren't in the table.  I'd recommend the following wording:  "List of institutions and the models that provided metrics from their CMIP6 models to IS-NES3, which represents a subset of the total of 124 models in CMIP6.  Also listed are HPC platform, and resolution used for the ATM and OCN components."  Now, proceed to comment below for a suggestion on the rest of the caption.

We updated the caption for Tab. 2. Now it reads:
List of Institutions and models that provided the metrics from their CMIP6 executions. Also listed are the HPC platform, and resolution used for the atmospheric and oceanic components}

Table 2:  In Table 1 the authors define "resol" as number of gridpoints, a singularly unhelpful metric when comparing models.  Fortunately, here in Table 2 they relate that metric to the more conventional lat-lon resolution.  However, I think in the table caption they should note something like, "Note "resol" in Table 1 is number of gridpoints. Here we define "ATM resol" as the horizontal resolution of the atmospheric model component in degrees of latitude and longitude, and "OCN resol" as the horizontal resolution of the ocean model component in degrees of latitude and longitude".

This paper builds on top of the previous definition for the metrics in Tab. 1 (Balaji. et al 2017), and the Resolution metric was defined as the number of gridpoints.
Nevertheless, we appreciate the feedback and we added a note for the reader when changing to degrees as in Tab. 2 caption, as we agree that it is more convenient for model comparisons. In the analysis section for the resolution, we also now use the degrees instead of the number of gridpoints.
Note, however, that we the authors have not designed these metrics, and therefore we collected the resolution following its definition, and only added the information in degrees for better understanding by the community.

Line 97:  Following the comment above, please add clarification here, something like, "…the categorization of low, medium and high resolutions in terms of latitude-longitude grid spacing. Thus, for the grouping…"
Line 99:  And again here, I recommend clarifying the wording as follows:  "…as low resolution with roughly 1 degree latitude-longitude grid spacing and up to …"

This part of the manuscript has been modified for clarity. While we maintain the gridpoint value, as it is required by the metric definition, we show the degrees threshold used for the categorisation:
"Most configurations have been categorised as low resolution and use up to 2.10E+07 grid-points in total, or no less than 0.7 degrees latitude-longitude grid spacing for any of the components (see Figure 1 and Table 2). On the other hand, only those experiments with an OCN/ATM resolution under 0.5 degree are treated as medium-high resolution configurations (see Figure 2).

Line 115:  Please define "complexity".  Does this mean number of components (e.g. atmosphere, ocean, sea ice, land ice, biogeochemistry, ocean ecosystem, cloud-aerosol interaction, etc.), number of parameterizations per component, or what?  This needs to be defined up front in order to make sense of the subsequent discussion.

We added an introductory paragraph at the beginning of the Complexity analysis section (3.2): The complexity of a coupled model, as defined in Table 1, is the number of  prognostic variables among all components. Here, "prognostic" refers to variables that the model directly predicts, such as temperature, atmospheric humidity, salinity, etc. In other words, variables that can be obtained directly as outcomes of the model.

Furthermore, we have also improved the definitions of other metrics that were not properly introduced before like the Complexity (section 3.2) Coupling cost (section 3.5) and Memory bloat (section 3.7)

---

## Author Comment (AC2)

Answer from authors to Reviewer 2 comments

**The computational and energy cost of simulation and storage for climate science: lessons from CMIP6**

**by**

Mario C. Acosta, Sergi Palomas, Stella Paronuzzi and Gladys Utrera

**General comments**

Acosta et al. present the results from the Computation Performance for Model Intercomparison Project (CPMIP). I believe this is the first time CPMIP has been run (but the relationship between CPMIP and other exercises isn't fully clear to me, see comments below). They demonstrate that there is a large range in computing requirements across modelling centres. They also include some nice analysis of what drives these differences in computing performance and the challenges associated with collecting their data. I think the paper provides some very interesting results that could be very helpful for the community. I also congratulate the authors on pulling together a paper like this (the research software engineer in our group has explicitly told us that they would never lead the writing of a paper so this effort to work in manuscript-land rather than the daily work of computing-land must be congratulated). However, the paper's key point is currently quite unclear to me and I think some areas are under-explored while others are given more time/space than is needed. I elaborate on my major and minor concerns below and recommend that the paper undergoes major revisions. As I said though, I congratule the authors on the effort they have put in to capture and share this information and hope that they have the energy to revise the paper and re-submit as it would be great to see it published in a revised form.

Before continuing, it should also be noted that I am not an HPC expert. I do a lot of analysis with ESM output but do not run them myself. As a result, it is also possible that most of my concerns arise simply because I am not the intended audience and, as such, I don't have enough background to meaningfully engage with the manuscript.

We extend our sincere gratitude to the reviewer for their feedback, which has been essential in improving the quality of our manuscript. Their observations have led to significant enhancements in several key areas. Firstly, we have restructured the abstract and introduction

to incorporate clear conclusions derived from our metrics analysis while also addressing the limitations of our work. Additionally, we've refined the introduction, focusing more directly on computational earth sciences to better align with the manuscript's content.

Moreover, in response to the reviewer's request for greater clarity, we have provided additional details regarding the project under discussion (IS-ENES3). To address concerns about metric uncertainty, we have dedicated a section 2.2 to explain this aspect more thoughtfully. Furthermore, we have redesigned Figure 5 to better suit the data being discussed as it was suggested and revised our carbon footprint comparisons (section 3.8) to include comparisons with the $CO_2$ equivalent of various activities within our earth science department, while also highlighting the overall benefits for climate science CMIP6 has.

As highlighted by the reviewer, certain metric definitions were poorly introduced, resulting in difficulties for readers in understanding their value and follow-up discussions. To address this concern, we enhanced the clarity of the definitions for the Complexity (section 3.2), Coupling cost (section 3.5), and Memory Bloat (section 3.7) by incorporating equations and offering clearer explanations within the corresponding sections of the manuscript as needed. Additionally, we are grateful for the reviewer's meticulous identification of technical issues listed at the end, which we have taken care of.

Please, find below detailed responses to the feedback provided by the reviewer.

**Major concerns**

**Point of the paper**

At present, the paper presents a collection of results. However, its key conclusion was not immediately clear to me.

We have rewritten the abstract and added a last paragraph in the introduction to highlight the main points, results achieved during the analysis, and conclusions of the paper, which should help the reader to understand what is our main focus.

For example, it isn't clear to me whether the computing performance of climate modelling centres is world leading, or whether there is much better computing performance seen in other fields and climate science is struggling to capture it for whatever reason (e.g. lack of funding for software/hardware/people, lack of expertise, lack of time for performance tuning). Put another way, are there lots of easy wins out there or are we already at the limit of computing power and improving from here will be incredibly difficult? Or does the data not allow us to have any insights into this question?

We have modified the second paragraph of the introduction. We have deleted some references to climate modelling and added Balaj 2015, Liu et al. 2013 and Bauer et al. 2021 to address the question by the reviewer.

This is particularly true in the quantitative sense: there are a lot of numbers in the paper, but I have no idea whether those numbers are a demonstration of excellence or a demonstration that a lot of performance is being left behind for whatever reason (although I must say that I have come away from the paper with a much better sense of what qualitatively drives computing performance and the trade-offs). I think this would be greatly helped if the authors were to include representative benchmarks next to each of their metrics (where possible, acknowledging that many of them are quite specialised to climate science hence comparisons from outside the field may not be possible). That would help know e.g. whether a memory bloat of 100 is reasonable in comparison to other computing programs or whether this much bloat is extraordinarily high.

The central discussion in this manuscript revolves around the comparison of the computational performance of various models in the CMIP6 project. Comparisons between these models, as presented in the paper, are inherently difficult due the complexity of climate codes, the wide range of models involved, possible implementations, programming strategies, and HPC infrastructures used. The collection of performance metrics presented is the first of its kind to date and is specially designed for models in climate science: it is meant to quantify computational patterns shared among these types of codes and therefore it is not representative of other application domains.
Given that there are no other measurements like the ones presented, comparisons are limited to what was collected and shown in the manuscript.
Nonetheless, thanks to the observations from the reviewer, we have refined the definition of some metrics to better clarify their significance and what the values shown actually mean. Moreover, we now state more clearly in the introduction the main points of the paper, preventing any potential misunderstanding. We have also clarified that the publication of this work is actually the first step ahead to make possible reasonable comparisons in the future, creating a first version of a database that did not exist until now in our field.

I recommend the authors think about the 1, 2 or 3 key points they want to get across in the paper, add those key conclusions to the abstract and then ensure that those key points come out clearly throughout the text. (Even if the answer is, the metric collection was so difficult and so variable across groups that we really can't make any strong conclusions about performance, that is still an answer that would be good to understand. Such an answer would then make clear that the key conclusion is that we need to get much better at metric collection before we can really identify where the next performance gains can come from.)

The abstract has been updated to highlight the main points and conclusions. Same for the last paragraph in the introduction.

**Which project is being discussed**

Highlighted in the Abstract and Introduction that the data comes primarily from the IS-ENES3 project.

It was quite unclear to me which project exactly is being presented here. The authors introduce the idea of CPMIP. However, IS-ENE3 is also introduced and it wasn't clear to me what the relationship between the two, if any, is. Similarly, the authors mention that this presents results for European groups (page 16, line 253). Are there are also results for non-European groups or are these not included because such groups aren't part of IS-ENE3?

Related other questions on this topic:

- Was IS-ENE3 responsible for ES-doc too? How does IS-ENE3 relate to the wider ESGF/CMIP effort?
- Is IS-ENE3 responsible for CMIP6 model result publishing? Or is that the remit of wider ESGF/CMIP efforts (or is IS-ENE3 the team that actually does the publishing within the wider ESGF/CMIP banner)?

ISENES3 is a consortium funded by a H2020 project. The main responsible behind ES-DOC is the University of Reading (integrand of IS-ENES3 as well).
IS-ENES3 consortium is composed of the most important weather and climate centres in Europe, which means that the partners from IS-ENES3 provided a significant contribution to the ESFG/CMIP effort, though they are just one part of the wider effort.
Moreover, IS-ENES3 was devoted to improving the infrastructure to make the ESGF/CMIP publication easier during the project life.
This has been clarified in the introduction of the manuscript.

- page 3, line 58: "As the reader can see, not all institutions provided the full collection of CPMIPs." Does this mean, "As the reader can see, not all institutions provided the full collection of CPMIP metrics." An institution can't provide CPMIP, it can only participate in CPMIP no?

This has been revised, now it states clearer that some institutions, within IS-ENES3, could not provide the full set of performance metrics (CPMIP) for all the experiments they conducted for CMIP6

**Uncertainty in measurements**

Computational benchmarking is notoriously difficult (results vary based on a whole bunch of factors which can be extremely difficult to control for). At the moment, there is no indication of

the uncertainty on these measurements at all. I know that doing this with high precision is probably impossible. However, even a rough sense of the order of magnitude of the uncertainty would be extremely helpful. For example, do you think that the modelling centre's reported results come with an uncertainty of e.g. +/- 1%, 10%, 100%, more? Even this rough indication would help the reader understand what they're looking at and how certain we are in measurement of different quantities (e.g. I suspect we are much more certain about resolution than we are about coupling cost) and how obvious the difference between modelling centres really is.

We created a new section (2.2) dedicated to this issue

**Minor concerns**

**Presentation**

The writing is generally pretty good, but there are definitely some rough patches which I think could be easily improved with another close read from the authors. I think the figures could be generally made clearer and more appealing as their key point doesn't really jump out at present (even using a package like seaborn in Python would provide an instant boost at almost zero time cost). I would also say that I found the use of extensive abbreviations extremely distracting and that the abbreviations made comprehending the manuscript significantly harder. I know that the authors are probably used to abbreviating things in code, but there aren't character restrictions in the manuscript format so I think just spelling things out is often the better option as it makes life for the reader much simpler (they can just read the text, they don't have to remember what 15 different abbreviations mean). If the abbreviations are to be kept, I think they need to be repeated (at minimum, referred to) in table captions so the reader has them available to them while reading the tables (or at least knows where to look to decode the abbrevations). Reviewer comment: Figures could be generally made clearer

Following the reviewer's suggestion that figures could be generally made clearer, we have improved Figure 5. Originally:

[Figure]

Following the reviewer's suggestion, we tried to use seaborn python library, including the medium-high resolution models and presenting the information as a scatter plot:

[Figure]

As observed, the data exhibit a large spread while the number of data points is relatively small. Although we acknowledge that this type of relationship is best shown as a scatter plot, we found it necessary to explore alternative visualisation methods for presenting the information effectively. Below, we present an improved version of the plot. Since there is no relation between consecutive experiments, we opted against using a line (as we did before). To incorporate the medium-high resolution experiments we use a logarithmic scale for both vertical axes. Furthermore, we avoided the use of unnecessary acronyms inside any of the plots.

[Figure]

Not only we think this is a better representation of the data, but also helped find relations between CHSY and Parallelisation that were more obscure before. We have also updated the analysis on the data shown consequently (Section 3.6).

Furthermore, we have avoided the use of acronyms in all manuscript images (except for experiment configuration and institution names).

Reviewer comment: abbreviations made comprehending the manuscript significantly harder.

We have checked that none of the tables or images use abbreviations without proper definition, as well as avoided using most of them.
We have also removed some abbreviations in the text. However, we have to keep the ones on Experiment and Institution names.

**Carbon footprint discussion**

Considering the climate cost of running these climate models is a good thing to do. However, I do think the current discussion is a bit one-sided for two reasons. Firstly, the carbon comparisons presented aren't that helpful in my opinion (driving a car non-stop for a year is hard to imagine). I think much more helpful comparisons would be to, e.g., the carbon associated with all the humans involved in CMIP. For example, how much carbon was required for the various meetings associated with CMIP over the years (or, perhaps a better example, the carbon associated with travel to the latest UNFCCC COP) and how does the computing carbon compare to this (my rough back of the envelope suggested the carbon from air travel for CMIP is probably similar order of magnitude, if not an order of magnitude more, than the carbon required for the computing, for COP, I have no idea but I would guess it is significantly more). (For this comparison, I think it's also worth noting that zero-emissions electricity is a well-understood technology, whereas zero-emissions travel is still very nascent.) (A different, perhaps more amusing and directly comparable, comparison point would be to compare the computational cost of CMIP's computing with the computational cost of running e.g. YouTube,

Google, NetFlix or Twitch.) Secondly, only presenting the costs without considering the benefits at all is one-sided. The benefits of CMIP are huge and shouldn't be ignored in any such conversation (particularly when the carbon cost of the computing is, in the scheme of things, relatively small both in absolute terms but also perhaps as a fraction of the wider CMIP effort).

We completely agree that there were other options to better compare in the carbon footprint discussion. We added, at the end of the Carbon Footprint section (3.8), the CO2 equivalents of our department activities (BSC), as well as the values provided by another climate research institution ( CERFACS). The new examples are more helpful comparisons to understand the work for our community. Additionally, CMIP6 benefits have been highlighted in the introduction and remembered in the carbon footprint discussion.

**Journal scope**

This paper is much more about computing performance than model development. I know that finding journals for exactly this topic is tricky, but my feeling reading this was that this paper may be better suited to a dedicated computing journal rather than a journal on Physics (obviously this is ultimately up to the editors though).

We decided to submit our manuscript to GMD based on several compelling reasons.
First of all, GMD is where the original paper about CPIMP performance metrics was first published by Balaji et al. 2017 (CPMIP: measurements of real computational performance of Earth system models in CMIP6, https://doi.org/10.5194/gmd-10-19-2017). Therefore, CPMIP studies have a referent in GMD.
Moreover, other institutions involved in our manuscript collection have published in this journal before, with some of the experiment configurations and models appearing in the journal's literature, some of which also have sections dedicated to computational performance. Below, we list some examples:
- Megann et al. 2014. GO5.0: the joint NERC–Met Office NEMO global ocean model for use in coupled and forced applications (https://doi.org/10.5194/gmd-7-1069-2014)
- Noije et al. 2021. EC-Earth3-AerChem: a global climate model with interactive aerosols and atmospheric chemistry participating in CMIP6 (https://doi.org/10.5194/gmd-14-5637-2021)
- Döscher et al. 2022. The EC-Earth3 Earth system model for the Coupled Model Intercomparison Project 6 (https://doi.org/10.5194/gmd-15-2973-2022)
- Mulcahy et al. 2020. Description and evaluation of aerosol in UKESM1 and HadGEM3-GC3.1 CMIP6 historical simulations (https://doi.org/10.5194/gmd-13-6383-2020)
- Seland et al. 2020. Overview of the Norwegian Earth System Model (NorESM2) and key climate response of CMIP6 DECK, historical, and scenario simulations (https://doi.org/10.5194/gmd-13-6165-2020)
- Stockhause et al. 2022. Twenty-five years of the IPCC Data Distribution Centre at the DKRZ and the Reference Data Archive for CMIP data

Furthermore, there are examples of manuscripts focusing on computing that have been accepted in this journal before:
- Yepes-Arbós et al. 2022. Evaluation and optimisation of the I/O scalability for the next generation of Earth system models: IFS CY43R3 and XIOS 2.0 integration as a case study (https://doi.org/10.5194/gmd-15-379-2022)
- Taewon Cho et al. 2022. Computationally efficiency methods for large-scale atmospheric inverse modeling: https://doi.org/10.5194/gmd-15-5547-2022
- Matsushima et al. 2023. Overcoming computational challenges to realize meter- to submeter-scale resolution in cloud simulations using the super-droplet method https://doi.org/10.5194/gmd-16-6211-2023
- Bishnu et al. 2023. Comparing the Performance of Julia on CPUs versus GPUs and Julia-MPI versus Fortran-MPI: a case study with MPAS-Ocean (Version 7.1). https://doi.org/10.5194/gmd-16-5539-2023

Therefore, considering the alignment of our work with the journal's focus and history, we believe GMD stands out as the most suitable journal for disseminating this work.

**Other questions/comments**

- Introduction: I think this could be re-formulated to focus more clearly on computing performance. There is some time spent on explaining the value of climate modelling more generally but I don't think that's really a point you have to prove for this paper so those sentences could be condensed/cut. I think this could make the start of the paper a bit more punchy and help to address the issue's related to making the paper's point clear (it's a paper about computing performance, not climate modelling importance). This may happen naturally as part of addressing the major concerns of course.

We have eliminated unnecessary text from the introduction. Instead, we have expanded upon the computational aspects and motivation of the work.

- page 2, line 26: Could you provide more explanation about why software evolves slower than hardware? This seems quite important/interesting as part of the bigger picture, yet it currently isn't explored at all (e.g. if there are computing gains that aren't being realised simply because too few research software engineers are employed, rather than because of any true technical barriers, then that is a powerful, insightful conclusion from this exercise and you authors are probably the best placed in the world to make such comments as you have done so much work in this space with multiple teams over the last years i.e. even though this evidence is only qualitative, it is still extremely powerful and the best collection we have right now)

Added more clarification on what challenges climate science has when it comes to code development in the introduction

- page 2, line 30: The comment, "bigger ensemble sizes to minimize the model's inherent uncertainty", seems too vague to me. Bigger ensemble sizes only lower some kinds of uncertainty, not all model uncertainty. I would suggest either making this line more specific or deleting it (I don't think you need it, the point that better computing performance will allow us to do more science and that is a good thing stands on its own pretty well)

Removed the part about ensemble sizes. Now it reads:
 Enhancing the performance of these models is crucial to boost the rate at which they can grow (in the resolution, complexity, and features represented) and to allow running faster and more cost-effective simulations which contribute to the advancement of climate research.

- Section 3.7: please put this memory bloat in context i.e. explicitly address whether this memory bloat is surprisingly high or in line with what computer programs normally use. A factor of 100 between memory used and ideal memory seems high at first glance, but maybe this is just how modern computers are given how many processes need to run in order to produce output (or I misunderstand the metric).

Added some clarification on the metric in section 3.7. Added some examples with halos. Added what the value should range and that future collections will help determine it with more exactitude

- It wasn't clear to me how exactly Cpl C is measured. If that could be clarified (or the fact that it can't currently be measured easily) that would be great. I think some of that happens in Section 4 so maybe this just needs to be foreshadowed when Cpl C is first introduced to help the reader understand why it is so vaguely defined compared to the other metrics.

Added the equation to better introduce the Coupling Cost metric in section 3.5.

- mixing of floating point and decimals throughout (both in text and in the tables) is distracting. Please pick one or the other and use it consistently throughout.

We have updated all manuscript tables addressing this issue

**Technical corrections**

page 2, line 37: "set of 12 performance metrics," --> "set of 12 performance metrics that define the".

The current phrasing seemed a bit odd to me.

Does this better capture what you mean? (Balaji defined CPMIP, here you now follow up?)

page 2, line 39: "they" --> "the performance metrics" (the text is too far from what you're referring too to use 'they' in my opinion, 'they' could also refer to ESMs which is what 'they' previously referred to)

page 2, line 40: "Tab." --> "Table". The abbreviation Tab. is quite unusual and was very distracting to me at least so I would just use the full word (it's not worth saving one character).

Same comment for all uses of Tab. throughout the paper.

page 2, line 41: "Sec." --> "Section". As above, I would just spell the word out to avoid distracting your reader with this unusual/unnecessary abbreviation.

Same comment for all use of Sec. throughout.

page 3, Table 1: 'cost in core-hours' --> 'cost, measured in core hours'

page 3, line 53: Who provided the support?

page 4, Table 2 caption: "Institution" --> "institution" (no need for capital here). Also what are units for atmosphere and ocean resolution?

Added in the caption that I use degrees for this table.

page 4, Table 2: make all numbers floating point or all decmials. Mixing 1/4 and 0.5 in the same column reads really weirdly.

page 5, Table 3 caption: "Institution" --> "institution" (no need for capital here)

page 5, Table 3: units for each column? What is 'Useful SY' (not in Table 1)?

Added this information in the table caption

page 5, Table 3: Adding a line at the bottom with the mean/median across all groups and the standard deviation would help to more quickly understand where each centre sits relative to its peers. For non-experts, if possible (and maybe it's not), it would also be very helpful to have some representative number so we can tell where performance is already good and where there

seems to be clear areas for improvement. (This comment can be made across all tables that present results from multiple groups)

The authors acknowledge that adding a bottom line with the mean/median and standard deviation would easily give a broad sense of the values for each metric.
However, due to the substantial diversity among models, the limited availability of data, the possible variability of the metrics, and the novelty of our collection (all of which is now explained in much detail thanks to the reviewer's comments), deriving meaningful summary statistics may not fully capture the nuances and complexities inherent in the comparisons.
Instead, during the analysis section, the authors focused in grouping models that are related in some way (e.g. using the same models but running in different platforms in table 7, separating low and medium-high resolution experiments in section 3.1, discerning queue time and queue time and system interruptions for the Actual SYPD calculation in section 3.4 , etc.).

page 5, Table 4: What are these metrics? What are their units? How do these compare to other machines around the world (like are climate centres using best in class machines or are there even more powerful ones around that are being used for other purposes, some help for the reader to understand the broader context would be great)?

At first, we collected the HPC machine metrics shown in Table 4 as we expected to be able to use the information for multi-platform comparison among similar experiments. While this ultimately fell outside the scope of our manuscript since we are focusing on climate science, we believe these metrics remain valuable. For instance, the Power Usage Effectiveness (PUE) is required for calculating the Carbon Footprint. Additionally, the readers can understand the hardware behind each CMIP experiment even though a platform comparison is not done.  We acknowledge that the caption of the table did not introduce the metrics adequately. The issue has been addressed by extending the caption.

.

page 9, line 193: 'to use' --> 'the use'

page 11, line 175: 'account up' --> 'account for up'

page 11, line 187: "Vegetation" --> "vegetation" (this random capitlisation occurs in quite a few places, please check)

Figure 5: I think this would be much better as a scatter plot. Also, do you have an equivalent figure for medium-high resolution models?

Table 11: What is a 'dwarf' in this context?

Deleted dwarf from Table 10. Now it reads:
Integrating the CPMIPs the High Performance Climate and Weather (HPCW, van Werkhoven et al., 2023) benchmark to evaluate the performance of the different machines used by the community.

Table 11: "for automize" --> "to automate"